# Structure and function of an *Arabidopsis thaliana* sulfate transporter

Lie Wang[1,2], Kehan Chen[1,2] & Ming Zhou ⬤ [1✉]

Plant sulfate transporters (SULTR) mediate absorption and distribution of sulfate ($SO_4^{2-}$) and are essential for plant growth; however, our understanding of their structures and functions remains inadequate. Here we present the structure of a SULTR from *Arabidopsis thaliana*, AtSULTR4;1, in complex with $SO_4^{2-}$ at an overall resolution of 2.8 Å. AtSULTR4;1 forms a homodimer and has a structural fold typical of the SLC26 family of anion transporters. The bound $SO_4^{2-}$ is coordinated by side-chain hydroxyls and backbone amides, and further stabilized electrostatically by the conserved Arg393 and two helix dipoles. Proton and $SO_4^{2-}$ are co-transported by AtSULTR4;1 and a proton gradient significantly enhances $SO_4^{2-}$ transport. Glu347, which is ~7 Å from the bound $SO_4^{2-}$, is required for H$^+$-driven transport. The cytosolic STAS domain interacts with transmembrane domains, and deletion of the STAS domain or mutations to the interface compromises dimer formation and reduces $SO_4^{2-}$ transport, suggesting a regulatory function of the STAS domain.

[1] Verna and Marrs McLean Department of Biochemistry and Molecular Biology, Baylor College of Medicine, Houston, TX, USA. [2] These authors contributed equally: Lie Wang, Kehan Chen. ✉email: mzhou@bcm.edu

Sulfur is an essential element for plants and it commonly exists as sulfate ions ($SO_4^{2-}$) in soil and water[1]. Higher plants have evolved a family of $SO_4^{2-}$ transporters, SULTRs, that mediate absorption of $SO_4^{2-}$ from the soil and its distribution to the entire organism[2–4]. SULTRs are composed of four subfamilies with approximately 60% amino acid sequence similarity. SULTR1 and 2 subfamilies are primarily responsible for $SO_4^{2-}$ uptake in the root and transport from root to shoot[5,6], SULTR3 facilitates $SO_4^{2-}$ uptake into chloroplasts[7,8], and SULTR4 is localized to vacuolar membranes, where they release the stored $SO_4^{2-}$ into the cytosol[9].

SULTRs are closely related to the mammalian solute carrier 26 family of transporters (SLC26), which consists of 11 members (SLC26A1–11)[10,11]. SLC26 family of proteins are anion exchangers or channels that mediate transport of inorganic anions such as $Cl^-$, $I^-$, $HCO_3^-$, and $SO_4^{2-}$, and small organic anions such as oxalate and formate[12–15]. Structures of a mammalian chloride channel (SLC26A9) were solved recently[16,17], and so were the structures of two bacterial homologs of SLC26, a bicarbonate transporter from cyanobacteria (BicA)[18], and a fumarate transporter from the bacterium *Deinococcus geothermalis* (SLC26Dg)[19]. These structures show a common architecture of a homodimeric assembly and that each monomer is composed of 14 transmembrane helices and a C-terminal cytosolic domain named Sulfate Transporter and Anti-Sigma factor antagonist (STAS) domain. While the general structural fold of plant SULTRs can be inferred from these structures because SULTRs are ~30% identical and ~47% similar to the mammalian SLC26A9, the existing structures are inadequate in guiding our understanding of substrate binding and transport in SULTRs due to differences in their substrates and mechanisms of transport.

In this work, we present results from structural and functional studies of the vacuolar SULTR4;1 isoform from *Arabidopsis thaliana*, AtSULTR4;1 (ref. [9]). We visualize the structure of AtSULTR4;1 with bound $SO_4^{2-}$ and demonstrate that AtSULTR4;1 is an $H^+/SO_4^{2-}$ symporter. We show that $H^+$ transport is likely mediated by a glutamate residue (Glu347) highly conserved among the plant SULTR family and that the cytosolic STAS domain modulates the transport process.

## Results

**$H^+$-dependent sulfate uptake by AtSULTR4;1.** Previous studies of SULTRs used cell-based assays and showed that $SO_4^{2-}$ uptake is significantly enhanced in lower pH and thus SULTRs were defined as $H^+/SO_4^{2-}$ symporters[2,4,20,21]. We expressed AtSULTR4;1 in insect cells and reconstituted the purified AtSULTR4;1 into liposomes to measure $SO_4^{2-}$ transport using $^{35}SO_4^{2-}$ (Methods, Fig. 1a, b). In the presence of a pH gradient of 7.5 inside and 5.5 outside of the liposomes, $^{35}SO_4^{2-}$ from the outside accumulates inside of the proteoliposomes over time and reaches a steady-state (Fig. 1b). In contrast, liposomes without AtSULTR4;1 only have less than one-twentieth of the $^{35}SO_4^{2-}$ and since the amount does not increase over time, it is likely caused by nonspecific absorption of $^{35}SO_4^{2-}$ on liposomes. We then measured transport activity by measuring the amount of $^{35}SO_4^{2-}$ uptake at the 5-min time point (Fig. 1c). In the absence of a pH gradient, $SO_4^{2-}$ uptake is similar to blank control in symmetrical pH 7.5 or 5.5, indicating that a pH gradient is required for $SO_4^{2-}$ transport and that a higher $H^+$ concentration alone is not sufficient to sustain $SO_4^{2-}$ uptake. The reliance of transport on a pH gradient is further demonstrated when we varied the pH gradient from 2.5 units ($pH_{in}/pH_{out}$ 7.5/5.0) to 0.5 units (7.5/7.0), and significant $SO_4^{2-}$ uptake occurs when the pH gradient is larger than 1.5 units. These results suggest that $SO_4^{2-}$ transport is obligatorily coupled to proton transport. When we

kept the pH gradient at 2 units while varying the external and internal pH, we found that $SO_4^{2-}$ uptake did not show a significant difference at $pH_{in}/pH_{out}$ of 7.0/5.0, 7.5/5.5, and 8.0/6.0, but the uptake is significantly reduced at 8.5/6.5, indicating that in addition to the requirement of a proton gradient, the transporter is also sensitive to proton concentrations (Fig. 1d). This property seems consistent with the physiological environment of AtSULTR4;1, because the pH inside the *Arabidopsis* vacuoles is typically 5.2–5.5 (refs. [22,23]).

We next estimated the stoichiometry of $H^+$ and $SO_4^{2-}$ by testing whether a membrane potential affects the transport process. We used a $K^+$ ionophore valinomycin in the presence of either symmetrical $K^+$ or a $K^+$ gradient to clamp the membrane potential to either 0 or +90 mV, and we found that $SO_4^{2-}$ uptake is not significantly different (Fig. 1e). This result suggests that the transport process is electroneutral and by inference, transport of one $SO_4^{2-}$ is accompanied by co-transport of two $H^+$. Combined, these results confirmed that AtSULTR4;1 is an $H^+/SO_4^{2-}$ symporter and provided information on the mechanism of transport.

We next examined substrate selectivity by testing whether $SO_4^{2-}$ uptake is affected in the presence of another anion at a 50-fold higher concentration (Fig. 1f). Among the anions tested, oxalate ($C_2O_4^{2-}$) almost completely inhibited $SO_4^{2-}$ uptake, while thiocyanate ($SCN^-$) also reduced $SO_4^{2-}$ uptake significantly but to a lesser degree. Acetate ($C_2H_3O_2^-$) and iodide ($I^-$) had a modest inhibitory effect on $SO_4^{2-}$ transport, while the slight increase of uptake was observed in the presence of bicarbonate ($HCO_3^-$) and dihydrophosphate ($H_2PO_4^-$). The larger divalent anion citrate ($C_6H_6O_7^{2-}$) had no inhibitory effect. Citrate is divalent under the experimental condition (pH 5.5) because its $pK_{a3}$ is ~6.4. These results show that the substrate-binding site of AtSULTR4;1 has a preference for certain substrates, and although some conclusions on substrate selectivity may be derived, more experiments on a broader selection substrate and with combination of binding and transport assays are required to reach a concrete conclusion on substrate selectivity.

**The overall structure of AtSULTR4;1.** We determined the structure of AtSULTR4;1 at pH 6.0 in the presence of $SO_4^{2-}$ by cryo-electron microscopy (cryo-EM) to an overall resolution of 2.8 Å. (Fig. 2a, Supplementary Fig. 1, Methods). The density map is of sufficient quality to allow de novo building of AtSULTR4;1 structure with all the transmembrane (TM) helices and the STAS domain (Supplementary Fig. 2). The final model consists of residues 70–644. Residues 1–69 and 645–685 (the N- and C-termini) were not resolved and the two regions are predicted to be unstructured based on their sequence.

AtSULTR4;1 forms a homodimer. Each monomer has 14 transmembrane helices (TM1–14) followed by a C-terminal STAS domain. The STAS domains are swapped between the neighboring subunits (Fig. 2b). The dimeric assembly and the overall structural fold of both the transmembrane domain and the STAS domain are conserved among AtSULTR4;1 and other members of the SLC26 family. AtSULTR4;1 structure is most closely aligned to that of the mouse SLC26A9 (PDB ID 6RTC [https://www.rcsb.org/structure/6RTC]) with an RMSD (Cα) of 1.6 Å for the transmembrane domain and 1.1 Å for the STAS domain (Supplementary Fig. 3). The structural similarity is expected as the two share 47% sequence similarity. However, there are features in the substrate-binding site and dimer interface that are unique to AtSULTR4;1. Since the STAS domain is known to be on the cytosolic side of the membrane, the membrane topology of the dimer is unambiguous, and this assignment of orientation is also consistent with the positive inside rule[24] (Supplementary Fig. 4).

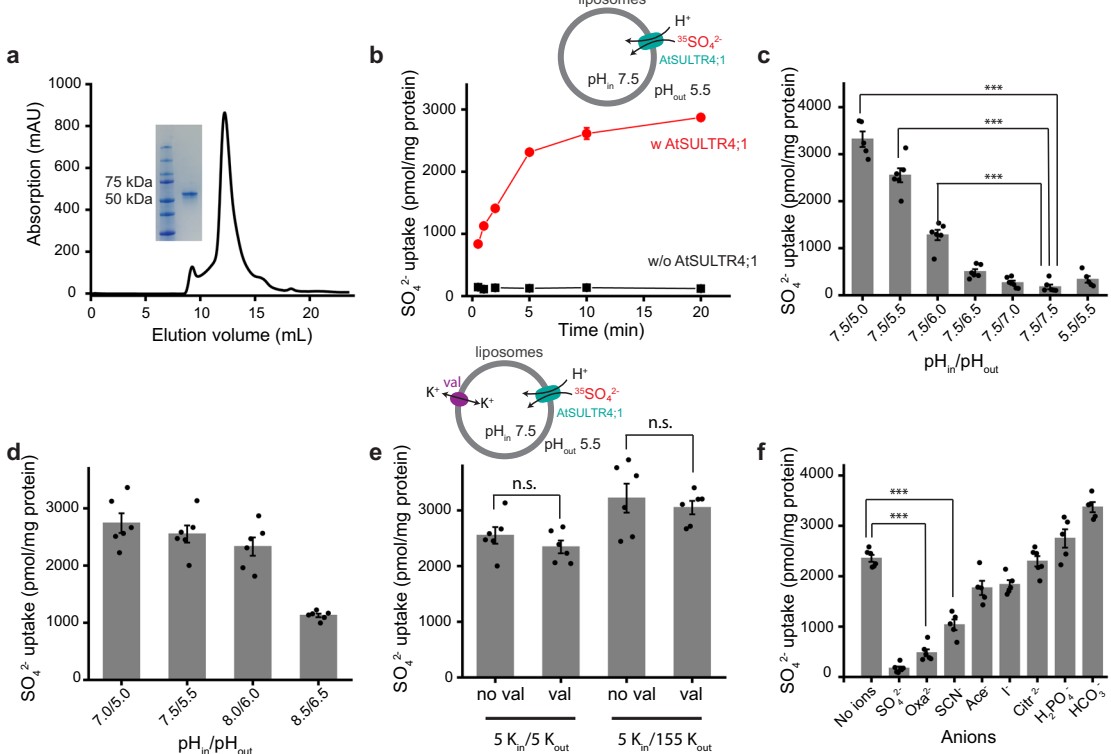

**Fig. 1 Functional characterization of AtSULTR4;1. a** Purification of AtSULTR4;1. The curve shows the size-exclusion chromatography profile of the purified protein, solubilized in detergent LMNG; the inset shows the SDS-PAGE profile of the purified protein. **b** Time-dependent $SO_4^{2-}$ transport by AtSULTR4;1. Radiolabeled $SO_4^{2-}$ is used to monitor sulfate influx. Red points represent proteoliposomes with reconstituted AtSULTR4;1, and black points represent liposomes without protein. Each data point is the average of 6 repeats from 2 batches of liposomes independently prepared. Error bars indicate the standard error of the mean (s.e.m.) of the data points. **c**, **d** pH-dependence of $SO_4^{2-}$ transport. The intra- and extra-vesicular pH values ($pH_{in}$ and $pH_{out}$, respectively) of each setup are indicated below the columns. For all column charts, a scatter plot of individual data points is overlaid onto each column. The height of each column represents the average of 5 or 6 repeats of experiments from 2 batches of liposomes independently prepared (in **c** and **d**, $n = 5$ for 5.5/5.5 and 7.5/5.0, and $n = 6$ for others). Error bars indicate s.e.m. of the average. Two-tailed Student's $t$ test was applied to selected data. *** indicates $p < 0.0001$. Exact $p$ values can be found in the Source Data file. **e** Voltage-dependence of $SO_4^{2-}$ transport. n.s. indicates that data are not significantly different. $n = 6$ for all columns. **f** Competition of $SO_4^{2-}$ transport by different anions. The pH values are 7.5/5.5 $pH_{in}/pH_{out}$ for these experiments. Anions are added at 50-fold the concentration of the radiolabeled $SO_4^{2-}$. $n = 5$ for $SCN^-$, $Ace^-$, $I^-$, $HPO_4^{2-}$, and $HCO_3^-$, and $n = 6$ for others.

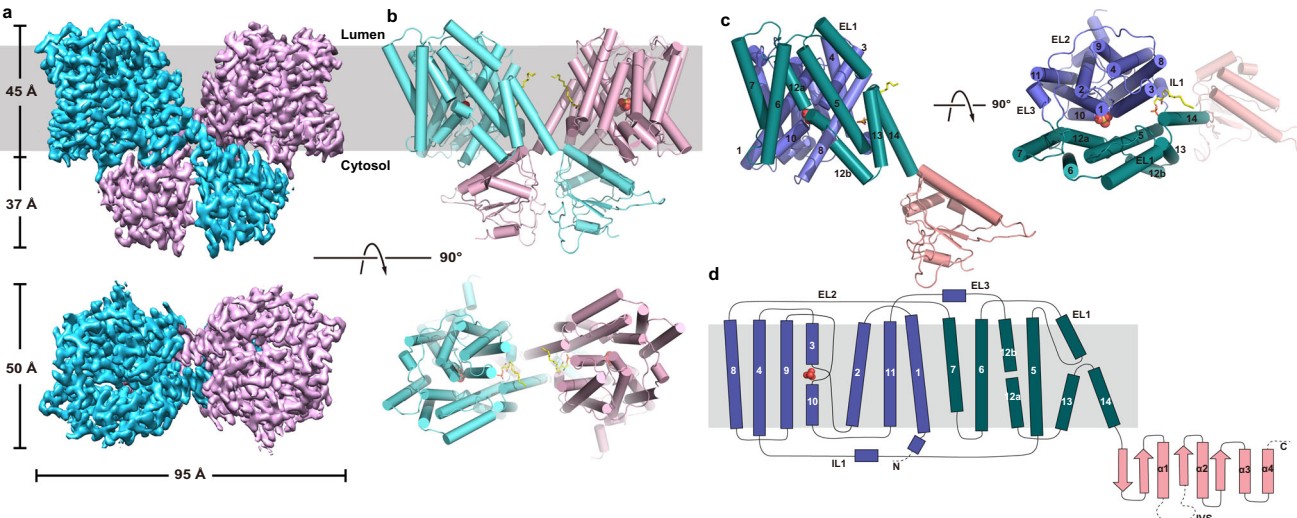

**Fig. 2 Structure of AtSULTR4;1. a**, **b** Electron density map (left) and cartoon representation (right) of the AtSULTR4;1 dimer as viewed from within the plane of the membrane (upper row), or the luminal side of the membrane (bottom row). Helices in the cartoon representation are shown as cylinders. **c** Cartoon representation of AtSULTR4;1 monomer in two orientations. The core domain, gate domain, and STAS domain are colored violet, teal, and pink, respectively. The bound $SO_4^{2-}$ is shown as spheres. **d** Topology of AtSULTR4;1.

The transmembrane helices of AtSULTR4;1 form a structural fold shared also by members of the SLC4 (bicarbonate transporter) and SLC23 (ascorbic acid transporter) families of transporters[16–19,25–29]. TM1–7 is related to TM8–14 by a pseudo twofold symmetry commonly seen in many other families of secondary solute transport proteins[30–32] (Supplementary Fig. 5). The 14 TM helices assemble into two distinct domains with a larger *core domain* formed by TM1–4 and 8–11, and a smaller *gate domain* by TM5–7 and TM12–14 (Fig. 2c, d). We use "core" and "gate" domains after the structure of UraA[27], which is a bacterial proton-dependent uracil transporter and is the first structure of this structural fold. The two domains are connected by three partially structured loops, two on the extracellular side (EL2 and 3), and one on the intracellular side (IL1). In the gate domain, TM13 and 14 are both significantly shorter than the rest of the helices, and a structured loop between TM5 and 6, EL1, folds on top of the two. In the core domain, TM3 and TM10 are half helices with the other halves in an extended conformation, and the two cross at approximately the center of the membrane. The crossover region forms a crater lined by elements from TM3 and 10, and residues from TM1 and TM8 as well (Fig. 3a, b). The crater is the substrate-binding site in several previous structures[18,27,28].

**SO$_4^{2-}$ and H$^+$ binding site**. In the density map of AtSULTR4;1, a clear nonprotein density is present at the crossover region of TM3 and 10. The density has a similar signal level to that of surrounding residues and fits well to a SO$_4^{2-}$ ion (Fig. 3a). The SO$_4^{2-}$ is located between the N-termini of TM3 and TM10, and is coordinated by side-chain and backbone atoms entirely from the core domain (Fig. 3b). The helix dipoles of TM3 and TM10 are oriented with their positive ends pointing to the bound SO$_4^{2-}$, providing the positive electrostatic potential to the bound SO$_4^{2-}$ (Fig. 3c). A conserved Arg393 from TM10 is ~5 Å away and although it does not make direct contact with the SO$_4^{2-}$, it likely provides positive electrostatic potential for the binding pocket. Sidechain

hydroxyls of Tyr116 and Ser392 are within hydrogen bond distance to SO$_4^{2-}$; and so are backbone amides of Ala153, Phe391, and Ser392. The position of the guanidinium group of Arg393 seems to be stabilized by interaction with Gln112 and Ser390 (Fig. 3d).

Sequence alignment across multiple SLC26 homologs shows that some of the residues at the SO$_4^{2-}$ binding site are highly conserved. For example, Gln112 is universally conserved in all eukaryotic homologs; the positive charge at Arg393 is preserved as arginine or lysine among all eukaryotic homologs except for SLC26A9, which is a known chloride channel; and an aromatic residue (Tyr or Phe) is present at Tyr116 (Supplementary Fig. 6). However, there is no clear pattern to indicate which residues are responsible for the selectivity of SO$_4^{2-}$ from other anions. To test how residues at the binding site affect transport, we mutated, one at a time, binding site residues Gln112, Tyr116, Ser392, and Arg393 to alanine, and found that the mutations have lost almost all transport activity (Fig. 3e, Supplementary Fig. 7). These results are consistent with the structure; however, more studies are required to determine how the residues at the binding site contribute to substrate binding and selectivity in SULTR.

We searched for protonatable side chains as potential carriers that mediate H$^+$ transport. A negatively charged residue, Glu347 on TM8, is ~7 Å away from the bound SO$_4^{2-}$. There is a strong density ~2.7 Å from the carboxylate of Glu347, 3.3 Å from the carbonyl oxygen of Gly389, and 3.7 Å from the hydroxyl of Thr388, and we assigned this density as a water molecule (Fig. 3d and Supplementary Fig. 2b). The density map also resolves two conformations of Phe391 at the N-terminal end of TM10 (Supplementary Fig. 2b). In the structures of the H$^+$-coupled fumarate transporter SLC26Dg and the Na$^+$-coupled bicarbonate transporter BicA, a glutamate or aspartate residue is also present at the equivalent position[18,19]. In BicA, Na$^+$ is thought to bridge the glutamate side chain and the bound bicarbonate anion[18]. However, this residue is not found in mammalian members of the SLC26 family except for SLC26A11 (Supplementary Fig. 6). We mutated Glu347 to examine its function. The E347A mutant has ~20% activity compared to wild type (WT), and the E347Q

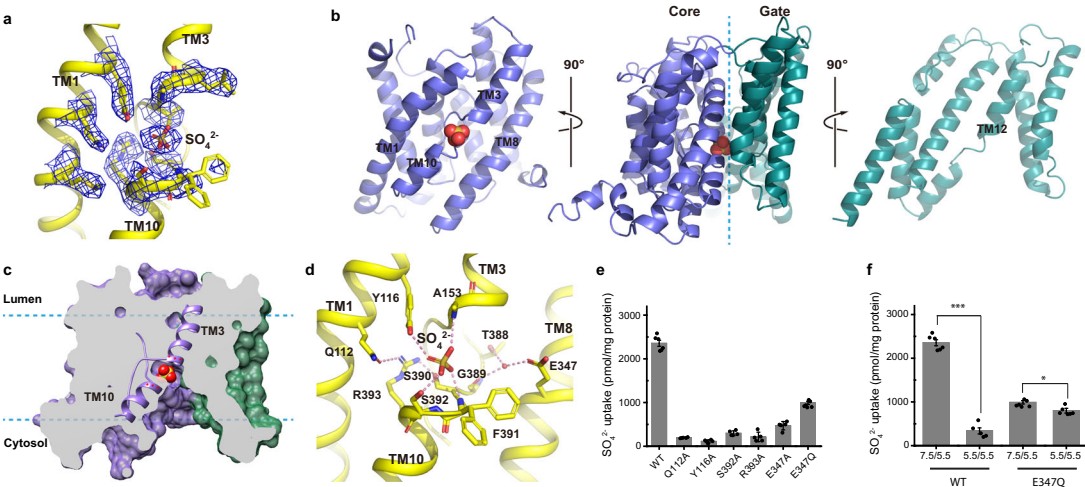

**Fig. 3 AtSULTR4;1 substrate binding site. a** Density of the bound SO$_4^{2-}$ and of residues nearby. Residues and SO$_4^{2-}$ are shown as sticks and the density is shown as blue mesh. **b** Cut-open view of the core and gate domains (violet and teal, respectively) of the TM domain of AtSULTR4;1. The two domains are rotated 90º as indicated to reveal their contact surface and the substrate-binding site in the core domain (left). **c** Cutaway view of the TM domain. Violet and teal surfaces indicate the core and gate domains, respectively. TM3 and TM10 are highlighted in cartoon representation. **d** Coordination of SO$_4^{2-}$ in the binding site. Direct interactions are marked with dashed lines. **e** Mutational studies of residues in the substrate-binding site. **f** pH dependence of AtSULTR4;1 WT and E347Q mutant. The pH$_{in}$/pH$_{out}$ conditions for each test are marked below the columns. The height of the columns represents the average of 5–6 repeats from 2 batches of liposomes independently prepared. $n = 5$ for Q112A, Y116A, S392A, E347A, and WT 5.5/5.5, and $n = 6$ for all others. Error bars indicate s.e.m. of the average. A two-tailed Student's $t$ test was applied to selected data. *** indicates $p < 0.0001$; * indicates $p < 0.01$. Exact $p$ values can be found in the Source Data file.

mutant retains ~40% transport activity (Fig. 3e). Interestingly, $SO_4^{2-}$ transport by the E347Q mutant is only slightly different between symmetrical pH and a pH gradient of 2 units (Fig. 3f). Combined, these results suggest that protonation and deprotonation of Glu347 are important for efficient transport of $SO_4^{2-}$ and for sensing a proton gradient.

**Dimer interface and the STAS domain.** The STAS domain is comprised of four α-helices (α1–4) and four β-strands (β1-4) (Fig. 2d), and the two STAS domains form a dimer. In mouse and human SLC26A9 structures, the first 30 residues of the N-terminus bind to the STAS domain and form part of the dimer interface[16,17]. However, in our structure, the N-terminus (residues 1–69) is not resolved and we do not observe densities at the corresponding position on the STAS domain (Supplementary Fig. 8). Although most of the dimer interface is contributed by the interactions between the two STAS domains, the domain-swapped arrangement in dimerization leads to the interaction of each STAS domain with the core and gate domains of the neighboring subunit (Fig. 4a, Supplementary Fig. 8).

In the current structure of AtSULTR4;1, both the bound $SO_4^{2-}$ and the putative proton sensing site Glu347 are solvent accessible only from the intracellular side, and thus the current structure is in an inward-facing conformation (Fig. 3c). Studies in the related SLC4 family of transporters suggest that transition between inward- and outward-facing conformations are achieved by a rigid-body motion between the core and gate domains[26,33,34]. If similar motions occur in the SLC26 family of proteins, the STAS domains could have a large impact on the motions of the core and gate domains. Interactions between the transmembrane domain and STAS domain are mainly electrostatic in nature. A number of charged residues are found in close proximity at the interface between the core and STAS domains (Fig. 4b–d). We made charge-reversal mutations to the residues on the core and STAS domains to test their effects on transport activity. Among the

mutants, K353E on TM8, and D547R, R550E, and K558E on α1 from the STAS domain all have significantly less transport activity (Fig. 4e). Notably, K353E, D547R, and R550E also have altered size-exclusion chromatography (SEC) profiles, consistent with weakened dimer assembly (Supplementary Fig. 7). The equivalent of Lys353 in human SLC26A4/Pendrin has a naturally occurring mutant K369E that is known to cause deafness[35]. These results suggest that the core–STAS interactions could modulate transport activity and stabilize the dimeric assembly. To further test the effect of the STAS domain, we truncated the entire STAS domain (residues 503–685) and term the construct AtSULTR4;1 ΔSTAS. The SEC profile of ΔSTAS showed a shifted elution volume corresponding to that of monomeric proteins (Supplementary Fig. 7). Interestingly, ΔSTAS retains a lower level of transport activity (Fig. 4e), indicating that the transmembrane domain is sufficient for $SO_4^{2-}$ transport.

## Discussion

Substrate binding sites in transporters from the SLC4/23/26 families have a highly conserved architecture. The binding site is composed of the following structural elements: TM3 and TM10 form a crossover and a crater-like pocket in the middle of the membrane; the crater is also lined by TM1 and TM8 on either side. TM3 and TM10 point their positive helix dipoles into the binding site; in all SLC26 family of transporters except for SLC26A9, a highly conserved Arg or Lys from TM10 provides additional positive potential; side-chain and backbone atoms from the N-termini of TM3 and 10 contribute to the coordination of the substrate; side chains from TM1 and in some structures these from TM8 contribute to the coordination of the substrate (Supplementary Fig. 9). In the bacterial UraA and fungal UapA, a glutamate residue on TM8 directly coordinates the substrate[27,28,36], while in AtSULTR4;1, Glu347 on TM8 is not close enough to make direct contact with the bound substrate, and our mutational studies suggest that Glu347 could mediate

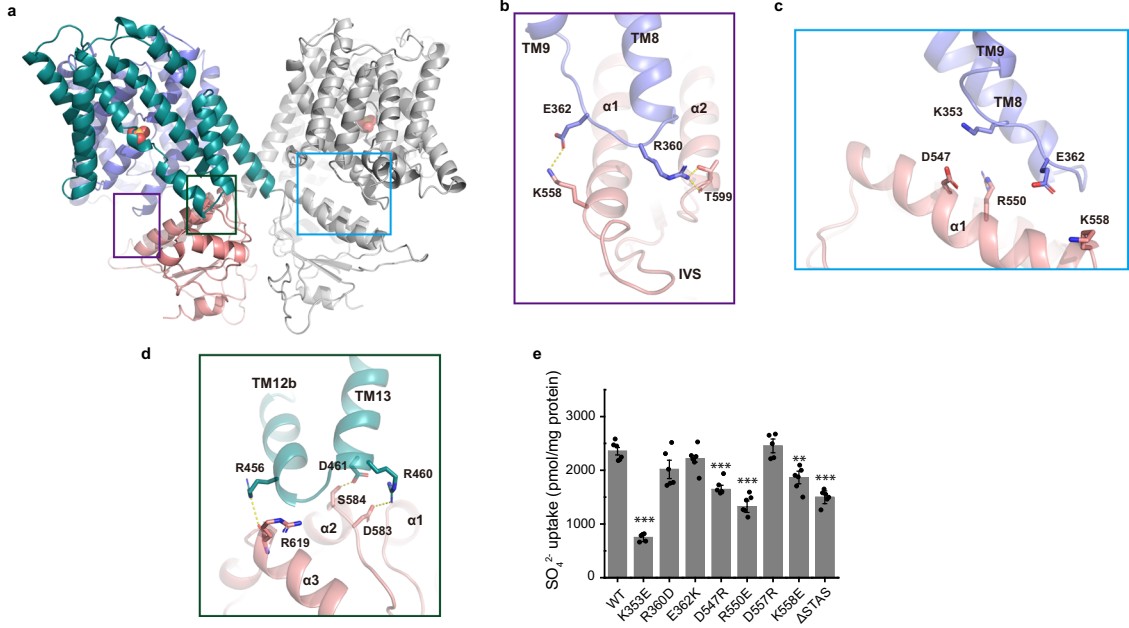

**Fig. 4 AtSULTR4;1 STAS and TM interaction. a** Structure of AtSULTR4;1. The TM domain of one protomer is violet and teal (for the core and gate domains, respectively), and the STAS domain from the other protomer is pink. The boxes are enlarged in (**b–d**) to show the STAS-core and STAS-gate interactions, respectively. **e** $SO_4^{2-}$ transport activities of charge-reversal mutants and the ΔSTAS construct. The height of the columns represents the average of 5–6 repeats from 2 batches of liposomes independently prepared. $n = 5$ for K353E, D547R, and D557R, and $n = 6$ for all others. Error bars indicate s.e.m. of the average. A two-tailed Student's $t$ test was applied to selected data. *** indicates $p < 0.0001$ compared to WT; ** indicates $p < 0.001$. Exact $p$ values can be found in the Source Data file.

proton binding and transport. Glu347 is conserved in all *A. thaliana* SULTR homologs, and likely has a similar role in other SULTR homologs, which are known to be pH-dependent[2,4,20]. The two bacterial homologs of SULTR/SLC26 also have a negative charge at the equivalent position of Glu347 (refs. [18,19]). BicA is Na$^+$-dependent and the aspartate is shown to coordinate a Na$^+$. SLC26Dg is H$^+$ dependent but it is not known if the glutamate mediates proton transport. No negatively charged residue is found in the corresponding position in human SLC26 isoforms, except for SLC26A11 (ref. [15]) (Supplementary Fig. 6). This clear distinction likely underlies the functional difference between plant and mammalian SULTR/SLC26 homologs: plant SULTR homologs are H$^+$/anion symporters, while most mammalian SLC26 homologs are anion exchangers or anion channels involving no co-transport of cations. We speculate that in AtSULTR4;1, protonation of Glu347 facilitates the initial binding of SO$_4^{2-}$ and its translocation, and deprotonation of Glu347 on the other side of the membrane encourages dissociation of the bound SO$_4^{2-}$. During this process, one H$^+$ is transported by Glu347 while the second H$^+$ through a pathway that likely involves water and other atoms of the protein. It is beyond the scope of the current study to define the second H$^+$ transport pathway.

The function of the STAS domain remains unresolved. Previous studies of the SLC26 family of transporters and their homologs showed that deletion of the STAS domain abolishes transport activity while mutations or truncations either reduce or abolish transport activity[18,19,37–40]. Mutations in the STAS domain of AtSULTR4;1 have reduced transport activity, while ΔSTAS retains some transport activity. We also found that the STAS domain contributes to the stabilization of the homodimer. Alignment of the current structure to that of mouse SLC26A9 (PDB ID 6RTC) provides some hints on the potential motion of the STAS domain. Although the isolated transmembrane and STAS domains of SLC26A9 align well with these of AtSULTR4;1 with an RMSD of 1.6 and 1.1 Å, respectively, aligning the full-length transporters gives an RMSD of 3.8 Å. One could account for the differences by allowing the two STAS domains to move simultaneously closer to the membrane (Supplementary Fig. 10). We speculate that the STAS domain may serve as a docking station for a regulatory protein or bind to small molecules that regulate transport activity.

The TM domains of the two subunits make very limited contact, and this feature is common to another SLC26 family of proteins (Supplementary Fig. 8)[16,17,41]. This is not unusual for membrane proteins with a substantial soluble domain as similar dimeric structures have been reported for the zinc transporters YiiP and ZnT8, and the cation-chloride transporter NKCC1[42–44]. The space between the two transmembrane domains would be filled with lipids, and several vestigial lipid densities can be seen between the neighboring subunits (Fig. 2c, Supplementary Fig. 2c, d). There is also a lipid-like molecule inserted between the core and gate domains.

In summary, the current study provides visualization of a transporter in the SULTR/SLC26 family in complex with its natural substrate, and defined mechanistic questions that will facilitate further experiments to understand the mechanism of substrate selectivity, H$^+$-coupled transport process, and regulation of transport by the STAS domain.

## Methods

**Cloning, expression, and purification of AtSulTR4;1.** The *A. thaliana* SULTR4;1 gene (NCBI accession number NM_121358.3) was codon-optimized and cloned into a modified pFastBac Dual expression vector[45] for production of baculovirus according to the Bac-to-Bac method (Thermo Fisher Scientific). P3 viruses were used to infect High Five (*Trichoplusia ni*) insect cells at a density of around $3 \times 10^6$ cells ml$^{-1}$, and

the infected cells were grown at 27 °C for 48–60 h before harvest. Cell membranes were prepared using a hypotonic/hypertonic wash protocol as previously described[45]. Briefly, cells were first lysed in a hypotonic buffer containing 10 mM 4-(2-hydroxyethyl)−1-piperazineethanesulfonic acid (HEPES) pH 7.5, 10 mM NaCl, and 2 mM β-mercaptoethanol (BME), 1 mM phenylmethylsulfonyl fluoride (PMSF), and 25 μg/ml DNase I. After ultracentrifugation at $55,000 \times g$ for 20 min, the pelleted cell membranes were resuspended in a hypertonic buffer containing 25 mM HEPES pH 7.5, 1 M NaCl, 2 mM BME, 1 mM PMSF, and 25 μg/ml DNase I, and were centrifuged again at $55,000 \times g$ for 20 min. Purified cell membrane pellets were flash-frozen in liquid nitrogen for further use.

Purified membranes were thawed and homogenized in 20 mM HEPES pH 7.5, 150 mM NaCl, and 2 mM BME, and then solubilized with 1.5% (w/v) lauryl maltose neopentyl glycol (LMNG, Anatrace) at 4 °C for 2 h. After solubilization, cell debris was removed by ultracentrifugation ($55,000 \times g$, 45 min, 4 °C), and AtSulTR4;1 was purified from the supernatant using a cobalt-based affinity resin (Talon, Clontech). The C-terminal His$_6$-tag was cleaved with tobacco etch virus protease at room temperature for 30 min. The protein was then concentrated to around 5 mg ml$^{-1}$ (Amicon 100 kDa cut-off, Millipore), and loaded onto a size-exclusion column (SRT-3C SEC-300, Sepax Technologies) equilibrated with 20 mM HEPES, 150 mM NaCl, 5 mM BME, and 0.01% (w/v) LMNG. For the sample used in cryo-EM, the size-exclusion column was equilibrated with 20 mM 2-(N-morpholino)ethanesulfonic acid (MES) pH 6.0, 150 mM Na$_2$SO$_4$, 5 mM BME, and 0.02% GDN (Anatrace).

AtSulTR4;1 mutant were generated using the QuikChange method (Stratagene) and the entire cDNA was sequenced to verify the mutation. Primer sequences are provided in the Source Data file. Mutants were expressed and purified following the same protocol as wild type.

**Cryo-EM sample preparation and data collection.** Cryo grids were prepared using the Thermo Fisher Vitrobot Mark IV. Quantifoil R1.2/1.3 Cu grids were glow-discharged in air for 15 s, 10 mA using the Pelco Easyglow. Concentrated AtSULTR4;1 (3.5 μl) was applied to each glow-discharged grid. After blotting with filter paper (Ted Pella, Prod. 47000-100) for 4.5 s, the grids were plunged into liquid ethane cooled with liquid nitrogen. For cryo-EM data collection, movie stacks were collected using SerialEM[46] on a Titan Krios at 300 kV with a Quantum energy filter (Gatan), at a nominal magnification of ×105,000 and with defocus values of −2.5 to −0.8 μm. A K3 Summit direct electron detector (Gatan) was paired with the microscope. Each stack was collected in the super-resolution mode with an exposing time of 0.175 s per frame for a total of 50 frames. The dose was about 50 e$^−$ per Å$^2$ for each stack. The stacks were motion-corrected with MotionCor2[47] and binned ($2 \times 2$) so that the pixel size was 1.08 Å. Dose weighting[48] was performed during motion correction, and the defocus values were estimated with Gctf[49].

**Cryo-EM data processing.** A total of 6,473,300 particles were automatically picked (RELION 3.1, refs. [50–52]) from 4660 images and imported into cryoSPARC[53]. Out of 200 two-dimensional (2D) classes, 11 (containing 340,063 particles) were selected for ab initio three-dimensional (3D) reconstruction, which produced one good class with recognizable structural features and three bad classes that did not have structural features. Both the good and bad classes were used as references in the heterogeneous refinement (cryoSPARC) and yielded a good class at 3.39 Å from 838,096 particles. Then nonuniform refinement (cryoSPARC) was performed with C2 symmetry and an adaptive solvent mask, which yielded a map with an overall resolution of 2.87 Å. Further CTF refinement yielded a map with an overall resolution of 2.75 Å. Resolutions were estimated using the gold-standard Fourier shell correlation with a 0.143 cut-off[54] and high-resolution noise substitution[55]. Local resolution was estimated using ResMap[56].

**Model building and refinement.** The structural model of AtSULTR4;1 was predicted and built based on the solved structure of mouse SLC26A9 (PDB 6RTC), and side chains were then adjusted based on the map. Model building was conducted in Coot[57]. Structural refinements were carried out in PHENIX in real space with secondary structure and geometry restraints[58]. The EMRinger Score was calculated as described[59].

**Proteoliposome preparation.** Soy polar extract lipids dissolved in chloroform (Avanti) were dried under a stream of Argon gas, and trace chloroform was removed with a vacuum. Dried lipids were rehydrated with 20 mM HEPES pH 7.5, 5 mM KCl, and 150 mM NaCl to a final concentration of 10 mg lipid per ml buffer. The rehydrated lipid mixture was sonicated to transparency and then went through three rounds of freezing and thawing. The liposomes were then extruded to homogeneity using a 400 nm filter membrane (NanoSizerTM Extruder, T&T Scientific Corporation) and were destabilized by the addition of 0.11% (w/v) Triton X-100. WT or mutant AtSULTR4;1 proteins were added at a 1:50 (w/w, protein:lipid) ratio. Detergent was removed by sequential addition of Bio-Beads SM-2 (BioRad) to the mixture. After detergent removal, the liposomes were aliquoted and flash-frozen in liquid nitrogen for further use.

**$^{35}SO_4^{2-}$ uptake assay**. Before the uptake experiments, liposome aliquots were extruded again to homogeneity. $SO_4^{2-}$ uptake was initiated by 10-fold dilution of liposomes (5 mg lipids per ml buffer) into the outside buffer containing 20 mM MES pH 5.5, 5 mM KCl, 150 mM NaCl, and 13.3 μM $^{35}SO_4^{2-}$. The radiolabeled $SO_4^{2-}$ was in the form of $Na_2SO_4$ (American Radiolabeled Chemicals, Inc), and the specific radioactivity was adjusted to 20 Ci mmol$^{-1}$ for all experiments. For experiments involving valinomycin, the salts in the outside buffer were either 5 mM KCl + 150 mM NaCl (0 mV) or 155 mM KCl (+90 mV). 200 nM valinomycin was added to clamp the membrane potential. Different buffers were used for experiments with different pH values: Na Citrate (pH 5.0), MES (pH 5.5–6.5), HEPES (pH 7.0–7.5), and Tris (pH 8.0–8.5). Reactions were stopped with the addition of ice-cold stopping buffer containing 20 mM Tris pH 7.5 and 150 mM LiCl and were filtered through 0.45 μm nitrocellulose filters (Millipore). The radioactivity retained on the filters was determined by liquid scintillation counting. A standard curve was plotted with known amounts of $^{35}SO_4^{2-}$ to convert counts per minute to pmol of $SO_4^{2-}$. Each data point represents the mean ± s.e.m values from five to six repeats from at least two batches of liposomes independently prepared.

## Data availability

The atomic coordinates of AtSULTR4;1 have been deposited in the PDB (http://www.rcsb.org) under the accession codes 7LHV. The corresponding electron microscopy maps have been deposited in the Electron Microscopy Data Bank (https://www.ebi.ac.uk/pdbe/emdb/) under the accession codes EMD-23351. Source data are provided with this paper.

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

## Acknowledgements

This work was supported by grants from NIH (DK122784, HL086392, and GM098878 to M.Z.), Cancer Prevention and Research Institute of Texas (R1223 to M.Z.). We acknowledge the cryo-EM core in Baylor College of Medicine for the support in grid preparation and screening. We are grateful to the Pacific Northwest cryo-EM national center for the support in data collection.

## Author contributions

M.Z., L.W., and K.C. conceived the project. L.W. and K.C. expressed and purified the proteins. L.W. solved the structure by cryo-EM, and K.C. characterized the function. All authors analyzed the data and wrote the paper.

## Competing interests

The authors declare no competing interests.
