## [Peer Review File · Nature Communications]

REVIEWER COMMENTS

Reviewer #1 (Remarks to the Author):

Sulfate transporter (SULTR) in plants mediates sulfate absorption from soil and transportation to the organisms. SULTRs comprises four subfamilies that share about 60% similarity with each other. The authors solved the 3D structure of a SULTR (SULTR4;1) from *Arabidopsis thaliana* at 2.8 Angstrom resolution, revealing the binding site for sulfate. The functional experiments prove the SULTR is an electroneutral proton/sulfate symporter. This work provides both structural and functional data to unveil the substrate binding mechanism of sulfate transporter and related homologs. The following are the points for this work.

1. lines 87-89, "while slight increase of uptake was observed in the presence of bicarbonate (HCO_3^-) and dihydrophosphate (H_2PO_4^-)." Why there was slight increase of uptake in the presence of HCO_3^- and H_2PO_4^- ? For H_2PO_4^- , it can become HPO_4^{2-} . What is the ratio between the dihydrophosphate ion and the hydrophosphate ion?
2. line 155, "The density map resolves two conformations of Phe391 at the N-terminal end of TM10 " Please show the density map for the two conformations of F391.
3. line 180-181, "There is also a lipid-like molecule inserted between the core and gate domains. " There is no displayed item for this statement. Please indicate and show the details.
4. line 181-182, "We speculate that lipid composition could have a significant effect on the transport process". Experimental data is needed to support this speculation/statement.
5. line 202-204, "In AtSULTR4;1, Glu347 on TM8 is not close enough to make direct contact with the bound substrate, but may affect substrate binding through its interaction with Phe319. " Is the interaction between E347 and F319 specific? What kind of interactions is it?
6. Figure 1f, there is a data for Citr2-. However, there is no description for this data in either main text or legend. Please describe and discuss the data for Citr2-.
7. Figure 3e, please change it more clearly to show the conservative residues.

Reviewer #2 (Remarks to the Author):

Wang et al present the cryo-EM structure of *Arabidopsis thaliana* sulfate transport in substrate bound conformation. The structure reveals the architecture of SULTRs which belong to the SLC26 family, and the sulfate binding site along with structural based analysis. The structure data is solid and worth to be published. I have a number of questions for the authors to take into consideration during revision.

1. The title: the sulfate binding site is revealed, but the mechanism of proton-coupled transport is not. i.e., the functional role of STAS domain remains unknown; the proton binding site is still putative. The authors may revise the title.
2. There lacks evidence supporting "the proton transport is mediated by Glu347" (abstract and Line154). At least some other mutation-based analysis should be carried out, i.e., E347Q.
3. A significant unsolved problem is the functional role of the STAS domain. Although the authors suggest that STAS is on the cytoplasmic side, they did not carry out functional analysis on functional

roles. Is it required for the transport activity? Do the interactions among the STAS and core/gate domain affect the transport activity (Line 169-172)? Are there any potential sulfate binding sites on STAS that allosterically regulate the activity (Line 197)?

4. Line 18, potential

5. Fig 3c is cited before 3a/b

6. L89, it is not reasonable to say StSULTR4.1 has a preference for divalent anions, as SCN⁻ also reduce the sulfate transport.

Reviewer #3 (Remarks to the Author):

This manuscript by Wang et al. describes a new experimental (cryo EM) structure of a SULTR plant sulfate transporter. Besides the high relevance of this group of transporters for plant physiology, they belong to the SLC26 family of anion transporters, ubiquitous throughout all organisms from bacteria to higher animals and plants. So far, there is little information on the molecular structures and transport mechanisms in this transport protein family, so new structures are highly welcome and important for the field.

The structure presented here is of high quality and plant SLC26 structures have not been reported so far. Also, robust functional assays are provided, defining the functional aspects of this protein, although the specific activity (pmol substrate/ μ g protein) is very low.

This said, I have two major criticisms:

First, the obtained structure is mostly confirmatory, agreeing well with the previously reported structures of bacterial (SLC26Dg) and animal (SLC26A9) SLC26 proteins. Also, most of the data are largely descriptive (e.g. STAS/core domain interaction), adding little to the understanding of the transport mechanism.

On the other hand, the authors identify a substrate in the binding pocket, shedding some light on the coordination of a divalent substrate.

Secondly, the authors suggest that Glu347 (in TM8 lining the binding site) mediates cotransport of H⁺, which would be highly relevant and different from the mammalian SLC26 transporters as illuminated by the recent cryo EM structure of SLC26A9. However, this conclusion is not well supported by the data. It is only a loss-of-function experiment (Glu347Ala) that supports this idea - which does not allow for conclusions about the mechanistic role of the residue. Moreover, the evidence for altered H⁺ cotransport is not convincing: whether the remaining small signal is different from background and thus indicates independence from H⁺ gradients remains unclear. Also, no attempt is made to reconcile the involvement of one titrable residue with the determined stoichiometry of 2 H⁺ per sulfate.

Other points:

- Throughout Introduction and Results, SULTR is presented as if it were not an SLC26 transporter, but more distantly related, which is confusing. Only in the Discussion SULTR is correctly classified as a bona

vide SLC26 (defined by sequence similarity and the presence of the STAS domain).

- The idea of STAS domains folding apart seems surprising and quite speculative: do the authors suggest that such a rearrangement is part of the transport cycle? Does that agree with the role of the STAS-STAS interface in dimerization?

- Figure 1e: Labeling is misleading. The voltages indicated are only clamped in the presence of valinomycin.

POINT-BY-POINT RESPONSE TO REVIEWER COMMENTS

Reviewer #1 (Remarks to the Author):

Sulfate transporter (SULTR) in plants mediates sulfate absorption from soil and transportation to the organisms. SULTRs comprise four subfamilies that share about 60% similarity with each other. The authors solved the 3D structure of a SULTR (SULTR4;1) from *Arabidopsis thaliana* at 2.8 Angstrom resolution, revealing the binding site for sulfate. The functional experiments prove the SULTR is an electroneutral proton/sulfate symporter. This work provides both structural and functional data to unveil the substrate binding mechanism of sulfate transporter and related homologs. The following are the points for this work.

1. lines 87-89, “while slight increase of uptake was observed in the presence of bicarbonate (HCO_3^-) and dihydrophosphate (H_2PO_4^-).” Why there was slight increase of uptake in the presence of HCO_3^- and H_2PO_4^- ? For H_2PO_4^- , it can become HPO_4^{2-} . What is the ratio between the dihydrophosphate ion and the hydrophosphate ion?

It is not clear why HCO_3^- enhances SO_4^{2-} transport. There may be some allosteric effects of this anion on the protein. It should be noted that at pH 5.5 HCO_3^- also exists as carbonic acid (H_2CO_3), which may further complicate the situation. The increase of uptake in the presence of H_2PO_4^- was not significantly larger than that of WT. Since the pKa for H_2PO_4^- is 7.2, at pH 5.5 outside the liposomes, more than 95% of the phosphate should be H_2PO_4^- .

2. line 155, “The density map resolves two conformations of Phe391 at the N-terminal end of TM10 ” Please show the density map for the two conformations of F391.

It is shown in Extended Figure 2b.

3. line 180-181, “There is also a lipid-like molecule inserted between the core and gate domains. ” There is no displayed item for this statement. Please indicate and show the details.

It is shown in Extended Figure 2c and 2d.

4. line 181-182, “We speculate that lipid composition could have a significant effect on the transport process”. Experimental data is needed to support this speculation/statement.

We do not yet have data to support this speculation, so we have removed this sentence.

5. line 202-204, “In AtSULTR4;1, Glu347 on TM8 is not close enough to make direct contact with the bound substrate, but may affect substrate binding through its interaction with Phe391. ” Is the interaction between E347 and F391 specific? What kind of interactions is it?

We thank the reviewer for pointing this out. The carbonyl group of the glutamate is $\sim 3.5 \text{ \AA}$ away from the closest carbon on the benzene ring when Phe391 is in one of the conformations. We

clarified our thoughts by the following statement: Thus, although Glu347 does not make direct contact with residues coordinating the bound SO_4^{2-} , it may affect substrate binding by its interactions with Phe391, Thr388 and Gly389.

6. Figure 1f, there is a data for Citr2-. However, there is no description for this data in either main text or legend. Please describe and discuss the data for Citr2-.

We added a description for citrate.

7. Figure 3e, please change it more clearly to show the conservative residues.

This panel has been replaced with Extended Data Figure 6, with more homologs added to show conservation.

Reviewer #2 (Remarks to the Author):

Wang et al present the cryo-EM structure of *Arabidopsis thaliana* sulfate transport in substrate bound conformation. The structure reveals the architecture of SULTRs which belong to the SLC26 family, and the sulfate binding site along with structural based analysis. The structure data is solid and worth to be published. I have a number of questions for the authors to take into consideration during revision.

1. The title: the sulfate binding site is revealed, but the mechanism of proton-coupled transport is not. i.e., the functional role of STAS domain remains unknown; the proton binding site is still putative. The authors may revise the title.

We changed the title to: Structure and function of an *Arabidopsis thaliana* sulfate transporter

2. There lacks evidence supporting “the proton transport is mediated by Glu347” (abstract and Line154). At least some other mutation-based analysis should be carried out, i.e., E347Q.

We made the E347Q mutation and found that it has reduced transport activity (~40%), and that the transport activity is less sensitive to a pH gradient.

3. A significant unsolved problem is the functional role of the STAS domain. Although the authors suggest that STAS is on the cytoplasmic side, they did not carry out functional analysis on functional roles. Is it required for the transport activity? Do the interactions among the STAS and core/gate domain affect the transport activity (Line 169-172)? Are there any potential sulfate binding sites on STAS that allosterically regulate the activity (Line 197)?

We made mutations to residues at the interface of transmembrane and STAS domains. 4 out of 7 mutants have significantly reduced transport activity. We also truncated the STAS domains and

found that it also has significantly reduced transport activity. In addition, some of the mutations become a mixture monomer and dimer after purification. While these results are consistent with a regulatory role for the STAS domain, we cannot deduce its function. We did not find additional densities on STAS domain that could be attributed to a sulfate ion.

4. Line 18, potential

Fixed.

5. Fig 3c is cited before 3a/b

Fixed.

6. L89, it is not reasonable to say StSULTR4.1 has a preference for divalent anions, as SCN⁻ also reduce the sulfate transport.

Fixed.

Reviewer #3 (Remarks to the Author):

This manuscript by Wang et al. describes a new experimental (cryo EM) structure of a SULTR plant sulfate transporter. Besides the high relevance of this group of transporters for plant physiology, they belong to the SLC26 family of anion transporters, ubiquitous throughout all organisms from bacteria to higher animals and plants. So far, there is little information on the molecular structures and transport mechanisms in this transport protein family, so new structures are highly welcome and important for the field.

The structure presented here is of high quality and plant SLC26 structures have not been reported so far. Also, robust functional assays are provided, defining the functional aspects of this protein, although the specific activity (pmol substrate/ μ g protein) is very low.

This said, I have two major criticisms:

First, the obtained structure is mostly confirmatory, agreeing well with the previously reported structures of bacterial (SLC26Dg) and animal (SLC26A9) SLC26 proteins. Also, most of the data are largely descriptive (e.g. STAS/core domain interaction), adding little to the understanding of the transport mechanism. On the other hand, the authors identify a substrate in the binding pocket, shedding some light on the coordination of a divalent substrate.

We agree that the structure of AtSULTR4;1 does not provide immediate revelations to the mechanism of transport or regulation by the STAS domain. Currently, all structures from the SLC26A family of transporters assume an inward facing conformation, suggesting that the outward facing conformation is less stable.

Secondly, the authors suggest that Glu347 (in TM8 lining the binding site) mediates cotransport of H⁺, which would be highly relevant and different from the mammalian SLC26 transporters as illuminated by the recent cryo EM structure of SLC26A9. However, this conclusion is not well supported by the data. It is only a loss-of-function experiment (Glu347Ala) that supports this idea - which does not allow for conclusions about the mechanistic role of the residue. Moreover, the evidence for altered H⁺ cotransport is not convincing: whether the remaining small signal is different from background and thus indicates independence from H⁺ gradients remains unclear. Also, no attempt is made to reconcile the involvement of one titratable residue with the determined stoichiometry of 2 H⁺ per sulfate.

We have made the E347Q mutation and found that it retains a lower transport activity and that its transport activity is less sensitive to a pH gradient. We interpret this result as evidence that protonation and deprotonation of E347 are required steps during the transport cycle.

We cannot reconcile the observation of 2 H⁺ per sulfate versus one titratable residue. There are no other titratable residues in the vicinity of substrate binding site. We speculate that perhaps protons do not have to be carried by titratable residues on the protein. For example, water molecules, other atoms on the protein, or the substrate itself can carry protons.

Other points:

- Throughout Introduction and Results, SULTR is presented as if it were not an SLC26 transporter, but more distantly related, which is confusing. Only in the Discussion SULTR is correctly classified as a bona fide SLC26 (defined by sequence similarity and the presence of the STAS domain).

We agree with the point and have now stated similarity of SULTR to SLC26 at the beginning of the manuscript.

- The idea of STAS domains folding apart seems surprising and quite speculative: do the authors suggest that such a rearrangement is part of the transport cycle? Does that agree with the role of the STAS-STAS interface in dimerization?

We do not intend to suggest that the STAS dimer falls apart and becomes two monomers during the transport cycle. We realized that the unintended effect of the cartoon and modified it to reflect the observation that the STAS domain is closer to the TM domain.

- Figure 1e: Labeling is misleading. The voltages indicated are only clamped in the presence of valinomycin.

Fixed. Thanks.

REVIEWERS' COMMENTS

Reviewer #1 (Remarks to the Author):

The authors revised the manuscript and addressed issues raised in previous round of review. The manuscript is suitable for acceptance.

Reviewer #2 (Remarks to the Author):

The authors respond to most of my questions and comments. I have no further questions.

Reviewer #3 (Remarks to the Author):

Overall, I am contended with the revisions of the manuscript.

There are just a few points left, improvement should be straightforward:

1. L 105. The authors define 3 extracellular and one intracellular loop. This is confusing. Obviously there are more extra- and intracellular TM-linkers (loops?). Please precisely define what constitutes an EL or IL.
2. L. 128: assumptions about the transport mechanism (and references to the literature) should better be moved to the Discussion.
3. L. 168: Again , this paragraph contains extensive speculation on the H⁺ transport mechanism, which, given the lack of experimental evidence (as correctly stated) should be treated in Discussion.

REVIEWERS' COMMENTS

Reviewer #1 (Remarks to the Author):

The authors revised the manuscript and addressed issues raised in previous round of review. The manuscript is suitable for acceptance.

Reviewer #2 (Remarks to the Author):

The authors respond to most of my questions and comments. I have no further questions.

Reviewer #3 (Remarks to the Author):

Overall, I am contended with the revisions of the manuscript.

There are just a few points left, improvement should be straightforward:

1. L 105. The authors define 3 extracellular and one intracellular loop. This is confusing. Obviously there are more extra- and intracellular TM-linkers (loops?). Please precisely define what constitutes an EL or IL.

We defined the extracellular and intracellular loops with the following description: “The two domains are connected by three partially structured loops, two on the extracellular side (EL2 and 3), and one on the intracellular side (IL1). In the gate domain, TM13 and 14 are both significantly shorter than the rest of the helices and a structured loop between TM5 and 6, EL1, folds on top of the two.”

2. L. 128: assumptions about the transport mechanism (and references to the literature) should better be moved to the Discussion.

This sentence is removed.

3. L. 168: Again, this paragraph contains extensive speculation on the H⁺ transport mechanism, which, given the lack of experimental evidence (as correctly stated) should be treated in Discussion.

Moved.